# ELAS: Efficient Pre-Training of Low-Rank Large Language Models via 2:4 Activation Sparsity

## Abstract

Large Language Models (LLMs) have achieved remarkable capabilities, but their immense computational demands during training remain a critical bottleneck for widespread adoption. Low-rank training has received attention in recent years due to its ability to significantly reduce training memory usage. Meanwhile, applying 2:4 structured sparsity to weights and activations to leverage NVIDIA GPU support for 2:4 structured sparse format has become a promising direction. However, existing low-rank methods often leave activation matrices in full-rank, which dominates memory consumption and limits throughput during large-batch training. Furthermore, directly applying sparsity to weights often leads to non-negligible performance degradation. To achieve efficient pre-training of LLMs, this paper proposes ELAS: **E**fficient pre-training of **L**ow-rank LLMs via 2:4 **A**ctivation **S**parsity, a novel framework for low-rank models via 2:4 activation sparsity. ELAS applies squared ReLU activation functions to the feed-forward networks in low-rank models and implements 2:4 structured sparsity on the activations after the squared ReLU operation. We evaluated ELAS through pre-training experiments on LLaMA models ranging from 60M to 1B parameters. The results demonstrate that ELAS maintains performance with minimal degradation after applying 2:4 activation sparsity, while achieving training and inference acceleration. Moreover, ELAS reduces activation memory overhead—particularly with large batch sizes. Code will be made available.

## 1 Introduction

Large language models (LLMs) have revolutionized numerous domains, from natural language processing and code generation to scientific discovery and multimodal understanding (Brown et al., 2020; Touvron et al., 2023; OpenAI, 2023). However, the computational demands of training these models have grown exponentially, with state-of-the-art systems requiring thousands of GPUs in compute resources (Patterson et al., 2021; Samsi et al., 2023). This has motivated extensive research into efficient pre-training methods that can reduce memory consumption and training costs while maintaining model performance. Among these approaches, low-rank training methods have emerged as a promising direction, offering significant reductions in memory usage through low-rank representations in weights or gradients. Notable examples include GaLore (Zhao et al., 2024), which projects gradients into low-rank subspaces to achieve memory efficiency while maintaining full-rank weights during training; LORO (Mo et al., 2025), which employs Riemannian optimization for genuinely low-rank weight training; CoLA (Liu et al., 2025), which introduces non-linear activations between low-rank weights to enhance the model's expressiveness; and LOST (Li et al., 2025), which combines low-rank and sparse structures through SVD-based initialization.

Recent literature has also explored the combination of low-rank and sparse structures to further enhance model expressivity. For instance, SLTrain (Han et al., 2024) proposes to construct low-rank model plus unstructured sparse model to improce the pre-training efficiency. LOST (Li et al., 2025) further construct low-rank and structured sparse weights with complementary SVD-based initialization, achieving improved training performance. While these approaches show promise, they primarily focus on weight-level sparsity. Directly applying sparsity on weight matrices can lead to a decline in model performance, as identifying the optimal sparse patterns and determining their proper initialization remains a challenge. Furthermore, the additional sparse components increases the overall architectural complexity. It is still an computing overhead for parameter efficient training.

Further, while these low-rank and sparse parameterizations reduce the memory footprint of weights and gradients, the resulting activation tensors remain full-rank, presenting a significant computational bottleneck. This limitation becomes particularly pronounced for large-scale models or large batch sizes, where activation storage and computation dominate the overall memory consumption and runtime complexity. The emergence of hardware-accelerated sparse computation offers a promising solution for efficient model training. Modern NVIDIA GPUs, starting from the Ampere architecture, provide native support for 2:4 structured sparse matrix multiplication. In this sparsity pattern, exactly 2 out of every 4 consecutive elements are non-zero to achieve 2× speedup compared with its dense equivalent (Mishra et al., 2021).

Recent efforts have shed light on the potential advantages of incorporating 2:4 sparsity for neural network acceleration. (Haziza et al., 2025) demonstrated that applying 2:4 sparsity to activations in models using Squared-ReLU can accelerate both training and inference of transformers, achieving up to 1.3× speedup in feed-forward networks. However, their approach focuses exclusively on full-rank models rather than low-rank training methods. As an orthogonal aspect, methods that apply 2:4 sparsity to weight matrices to gain pre-training efficiency generally sacrifice model accuracy. For example, (Hu et al., 2024b) proposed S-STE with a continuous pruning function to achieve 2:4 sparsity on weight matrices during pre-training. Despite S-STE's promising results, directly pruning weight matrices with 2:4 sparsity faces fundamental limitations. This is because weight matrices lack the natural sparsity characteristics that emerge in activations following non-linear transformations.

To address these challenges, we propose a novel low-rank plus sparse framework where sparsity is applied to activations rather than weights. We introduce ELAS (Efficient pre-training of Low-rank LLMs via 2:4 Activation Sparsity), which combines the parameter efficiency of low-rank training with the computational acceleration of structured activation sparsity. ELAS first applies low-rank decomposition to weight matrices to reduce parameter memory requirements. Then, ELAS employs Squared-ReLU activation functions (So et al., 2021) to naturally induce high sparsity levels in activations and applies magnitude-based 2:4 structured sparsity to these activations during forward passes. To maintain gradient flow during backpropagation, ELAS utilizes straight-through estimation to handle the non-differentiable sparsification operation. Our key contributions are:

① We propose ELAS, a novel low-rank plus sparse framework that seamlessly integrates low-rank weight training with 2:4 structured activation sparsity. Our method introduces a magnitude-based 2:4 sparsification algorithm for activations combined with straight-through estimation for gradient flow, enabling efficient sparse-dense matrix multiplication acceleration while maintaining low-rank training benefits. By shifting the sparse structure to activations, the computational overhead of the auxiliary sparse weight modules are eliminated. ELAS maintains full expressibility of low-rank factors while utilizing activation sparsity for hardware-native acceleration.

② We demonstrate that applying 2:4 sparsity to activations rather than weights in low-rank models provides substantial acceleration for inference while reducing memory consumption, particularly with large batch sizes. By employing Squared-ReLU activation functions (So et al., 2021), we naturally induce high sparsity levels in activations (84-98% sparse after warmup) (Haziza et al., 2025). ELAS minimizes performance loss when applying the 2:4 sparsity.

③ We validate ELAS through comprehensive experiments on LLaMA models ranging from 60M to 1B parameters on the C4 dataset (Raffel et al., 2020), demonstrating that our method maintains competitive perplexity compared to full-rank baselines while achieving meaningful speedups and activation memory reduction.

## 2 Related work

### 2.1 Low-rank Pre-training

The development of memory and parameter-efficient training methods for LLMs has gained lots of attention over recent years. Early foundational work by (Khodak et al., 2021) established spectral initialization and Frobenius decay for factorized neural layers. Additionally, they demonstrated that proper initialization using SVD and regularizing matrix products could make factorized networks competitive with full-rank counterparts. (Kamalakara et al., 2022) provided an extensive empirical study of low-rank training, and showed that pre-training offers significant speedups in language models but limited gains in vision tasks. (Saada & Tanner, 2023) extended edge-of-chaos initialization theory to

low-rank networks, revealing increased training variability as a fundamental limitation. (Lialin et al., 2023) introduced high-rank training through sequences of low-rank updates with periodic merging, achieving substantial computational savings. (Zhang et al., 2024) demonstrated that low-rank parametrization in Transformers exhibits steeper scaling curves with significant efficiency gains. (Zhao et al., 2024) took a different approach by projecting gradients into low-rank subspaces rather than constraining parameters, enabling 7B model training on consumer hardware. Recent advances include (Mo et al., 2025) which applied Riemannian optimization principles to ensure optimal gradient descent paths in low-rank training, (Liu et al., 2025) which replaced linear layers with low-rank compositions separated by non-linear activations, (Han et al., 2024) which combined low-rank and sparse factorization with random support strategies, and (Zhang & Papyan, 2024) which achieved state-of-the-art post-training compression through outlier-aware sparse and low-rank decomposition. Building upon these developments, (Li et al., 2025) presented a novel approach that uses SVD to initialize complementary low-rank and sparse components, achieving superior performance while maintaining significant computational and memory efficiency compared to existing methods.

## 2.2 Sparse training and N:M structured sparsity

The application of sparse training to LLMs initially focused on post-training pruning methods before advancing to more sophisticated training processes. (Sun et al., 2023) introduced a simple and effective pruning approach that evaluates weight importance by multiplying weight magnitudes with corresponding input activation norms. It achieved competitive results with SparseGPT (Frantar & Alistarh, 2023) while being faster. Outlier Weighed Layerwise Sparsity (Yin et al., 2024) changed the conventional practice of uniform layerwise sparsity by proposing non-uniform sparsity ratios based on each layer's outlier distribution.

Various approaches have emerged for training structured sparse networks to harness GPU acceleration capabilities. (Zhou et al., 2021) trained sparse networks with N:M patterns, where the Sparse-Refined Straight-Through Estimator and the Sparse Architecture Divergence metric are introduced to stabilize sparse structure updates during training. However, early attempts at structured sparsity struggled to train reliably with modern optimizers. (Lu et al., 2023) proposed a two-phase training approach with reliable variance estimates to address the incompatibility of existing sparse training techniques with the Adam optimizer. (Hu et al., 2024a) proposed to accelerate transformer pre-training using 2:4 structured sparsity. It introduced a transformer-specific masked decay and practical acceleration methods that achieved speedups of up to 1.2x when training GPT-2 models. Subsequently, (Hu et al., 2024b) addressed optimization difficulties in 2:4 sparse training by proposing a continuous soft-thresholding pruning function that maintains 2:4 sparsity while enabling smooth optimization.

Several approaches emerged to achieve bidirectional acceleration, i.e., forward and backward passes, during structured sparse neural network training. (Zhou et al., 2021) introduce N:M transposable fine-grained sparsity masks to formulate the problem as a minimum-cost flow optimization with a 2-approximation algorithm. Alternative solutions included (Zhang et al., 2023), which proposed using separate sparse masks for forward and backward directions rather than requiring a single transposable mask. The exploration of activation functions has played a crucial role in enabling effective sparse training. GLU variants (Shazeer, 2020) demonstrated that gating mechanisms, particularly GEGLU and SwiGLU, could enhance Transformer performance across various tasks by replacing standard feed-forward layers with gated architectures. (So et al., 2021) further validated the effectiveness of squared ReLU activations. It revealed that squared ReLU in feed-forward blocks enabled significant training cost reductions while maintaining or improving performance. Most recently, researchers also explored the natural sparsity properties of activation functions for training acceleration. (Haziza et al., 2025) applied structured sparsity patterns to model activations rather than weights. This approach leverages the intrinsic sparsity naturally found in Squared-ReLU activation functions.

## 2.3 Low-rank and sparse training

To address the limited expressiveness inherent in purely low-rank approximations, recent research has explored the combination of low-rank and sparse structures. This approach is rooted in Robust Principal Component Analysis (RPCA) (Candès et al., 2011), which seeks to decompose a matrix into a low-rank component capturing global structure and a sparse component representing localized outliers or residuals.

In the context of LLMs, low-rank and sparse decomposition has demonstrated significant potential for model compression and efficient fine-tuning. For instance, Zhang & Papyan (2024) proposed OATS, which explicitly approximates weight matrices as the sum of a sparse matrix and a low-rank matrix. To minimize reconstruction error, OATS applies

an alternating thresholding algorithm that iterates between singular-value thresholding and hard-thresholding, while preserving critical outlier features by scaling weights according to the second moment of input embeddings. Makni et al. (2025) introduced HASSLE-free, a unified optimization framework for low-rank and sparse decomposition. It treats the compression as a formal alternating-minimization problem that directly minimizes the layer-wise reconstruction error using the full Hessian. HASSLE-free integrates structured sparsity into the sparse component to leverage hardware acceleration. It utilizes diagonal scaling to maintain numerical stability in ill-conditioned transformer layers.

To avoid the prohibitive cost of training full-rank models, recent research explores low-rank and sparse architectures for pre-training. Pixelated Butterfly (Chen et al., 2022) optimizes over a continuous superset of block-aligned butterfly matrices. By adding a low-rank component to the flat butterfly structure, it compensates for the loss of expressiveness in sparse factors, achieving a significant speedup during training. Han et al. (2024) introduced SLTrain, which parameterizes layers as the sum of a low-rank matrix and an unstructured sparse matrix. However, SLTrain relies on standard Kaiming initialization for the low-rank part and random support for the sparse part, which can lead to suboptimal performance. The use of unstructured sparsity in SLTrain limits actual hardware speedup and incurs a memory overhead. To solve these limitations, Li et al. (2025) developed LOST, which utilizes SVD-based initialization to partition the weights into complementary low-rank and structured sparse components. By leveraging structured sparse weights, LOST achieves improved efficiency compared to SLTrain.

However, both SLTrain and LOST focus on sparsifying the weight matrices which increases model complexity. In contrast, our proposed ELAS shifts the focus to activation sparsity within a low-rank framework. This pattern eliminates auxiliary sparse weight modules while capturing the natural sparsity emerging during the non-linear transformations of the forward pass.

## 3  Methodology

In this section, we introduce ELAS that combines low-rank weights with activation 2:4 sparsity to achieve efficient training. ELAS integrates two key components: low-rank weight matrices and 2:4 sparsification of forward activations to enable hardware-accelerated sparse matrix multiplication. Figure 1 shows the forward process of the ELAS's feed-forward network. The design rationale of ELAS is to utilize the low-rank weights to save gradient and optimizer memory while adopting 2:4 structured sparsity on the activations to accelerate the forward/backward passes and save activation memory. This allows ELAS to overcome the limitations of existing low-rank plus sparse methods.

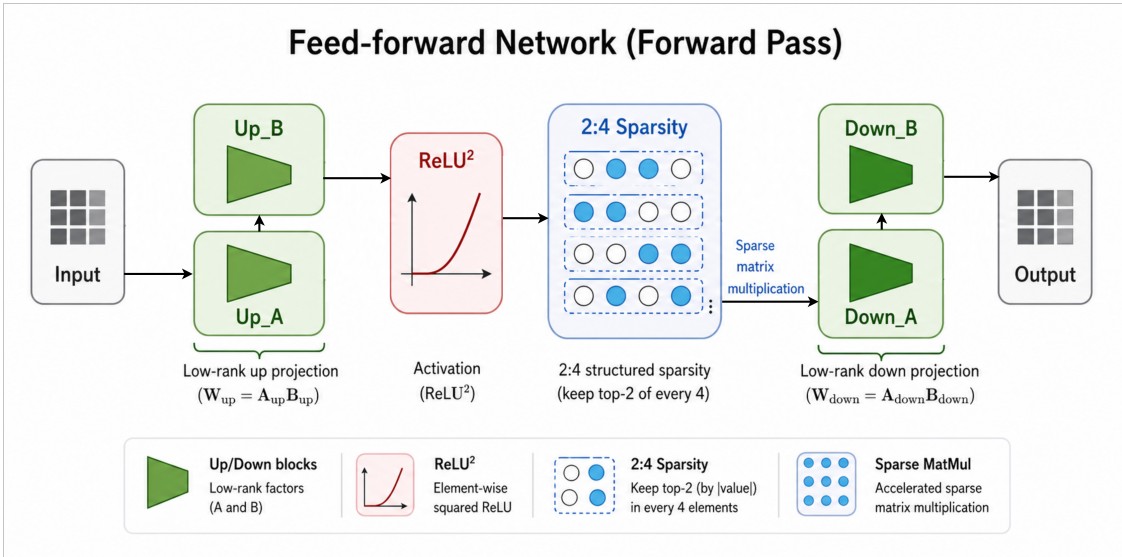

Figure 1: Feed-forward network architecture of the ELAS. The input is first multiplied by the low-rank matrices of the up projection layer, then passes through the $\text{ReLU}^2$ activation function. The activation is applied with 2:4 structured sparsity and then multiplied with the low-rank matrix of the down layer using sparse matrix multiplication to obtain the output of the FFN layer.

### 3.1 Low-Rank Training Framework

We adopt the LORO (Mo et al., 2025) framework as the foundation of our method. The core principle of LORO is to construct models with low-rank weight matrices while maintaining training stability through orthogonal gradients update constraints.

For a given linear layer with weight matrix $\mathbf{W} \in \mathbb{R}^{d_{out} \times d_{in}}$, LORO replaces it with two low-rank matrices:

$$\mathbf{W} = \mathbf{AB} \tag{1}$$

where $\mathbf{A} \in \mathbb{R}^{d_{out} \times r}$ and $\mathbf{B} \in \mathbb{R}^{r \times d_{in}}$ with rank $r \ll \min(d_{out}, d_{in})$.

The LORO optimization procedure begins with initialization, where matrices $\mathbf{A}$ and $\mathbf{B}$ are initialized using Xavier initialization. LORO alternates between approximate Riemannian updates (most steps) and exact Riemannian updates (every $K$ steps). The exact updates involve SVD-based reparameterization that shifts the optimization to new subspaces, requiring optimizer state refresh to maintain training stability

We apply LORO optimization to both attention and MLP layers, with configurable ranks $r_{attn}$ and $r_{mlp}$ for low-rank layer. In our experiments, we set $r_{attn} = r_{mlp} = 256$ to balance parameter efficiency with model expressiveness.

We specifically choose LORO over other low-rank methods such as CoLA (Liu et al., 2025) that adds activation functions between low-rank factors, or LOST (Li et al., 2025) that incorporates additional sparse matrices for a practical reason: LORO maintains the cleanest inference pathway. After training completion, there is no additional sparse matrix or activation function between low-rank matrices, resulting in the fastest inference speed. This is particularly important for applications where both training efficiency and deployment performance are priorities.

### 3.2 Model Architecture Modification

Following (Haziza et al., 2025), we modify the standard LLaMA architecture by replacing the SwiGLU activation function in MLP layers with a simpler FFN structure using $\text{ReLU}^2$ activation. This architectural change is motivated by the natural sparsity properties of $\text{ReLU}^2$, which yields high activation sparsity levels that are well-suited for 2:4 structured sparsity applications. Additionally, by removing the gating mechanism present in SwiGLU, this simplified structure reduces computational overhead while maintaining nonlinearity for model expressiveness.

The modified architecture follows:

$$\text{MLP}(\mathbf{x}) = \mathbf{W}_{down} \cdot \text{ReLU}^2(\mathbf{W}_{up}\mathbf{x}) \tag{2}$$

where $\text{ReLU}^2(\mathbf{z}) = (\max(0, \mathbf{z}))^2$. This modification provides a simplified structure by removing the gating mechanism, which reduces computational overhead while maintaining nonlinearity. It induces sparsity since $\text{ReLU}^2$ naturally produces high activation sparsity, making it suitable for structured sparsity.

### 3.3 2:4 Activation Sparsity

To leverage the natural sparsity patterns inherent in $\text{ReLU}^2$ activations while enabling hardware acceleration, we apply structured 2:4 sparsity to forward activations of low-rank layers. The 2:4 sparsity pattern requires that out of every consecutive 4 elements, at least 2 are zero, enabling efficient sparse matrix multiplication on supported GPUs.

#### 3.3.1 Forward Pass Sparsification

For activation tensor $\mathbf{z} \in \mathbb{R}^{M \times N}$ after $\text{ReLU}^2$, we apply 2:4 structured sparsity element-wise along the feature dimension. As shown Algorithm 1, each row is partitioned into non-overlapping groups of four consecutive elements; within each group, we retain the two entries with the largest absolute values and set the other two to zero.

The sparsification operation is implemented using efficient Triton kernels for GPU acceleration:

$$\text{sparsify}_{2:4}(\mathbf{z}) = \text{mask}_{top2}(\mathbf{z}) \odot \mathbf{z} \tag{3}$$

where $\text{mask}_{top2}(\mathbf{z})$ generates a binary mask that preserves only the top-2 elements by absolute value in each group of 4.

---

**Algorithm 1:** 2:4 Activation Sparsification

---

1: **Input:** Activation tensor $\mathbf{z} \in \mathbb{R}^{M \times N}$
2: **Output:** Sparsified tensor $\mathbf{z}_{sparse} \in \mathbb{R}^{M \times N}$
3: **for** $i = 1$ to $M$ **do**
4:   **for** $j = 1$ to $N/4$ **do**
5:     **group** $= \mathbf{z}[i, 4j - 3 : 4j]$ {Extract group of 4 elements}
6:     **abs_group** $= |\mathbf{group}|$ {Compute absolute values}
7:     Find indices of top-2 largest values in **abs_group**
8:     Set remaining 2 elements to zero
9:     $\mathbf{z}_{sparse}[i, 4j - 3 : 4j] = \mathbf{group}_{masked}$
10:   **end for**
11: **end for**

---

### 3.3.2 Backward Pass with Straight-Through Estimator

During backpropagation, we employ the straight-through estimator (STE) to process the non-differentiable sparsification operation. The STE allows gradients to flow through the sparsified activations by treating the sparsification operation as an identity function during the backward pass:

$$\frac{\partial \mathcal{L}}{\partial \mathbf{z}} = \frac{\partial \mathcal{L}}{\partial \mathbf{z}_{sparse}} \tag{4}$$

This approximation allows gradient-based optimization to continue despite the discrete nature of the sparsification operation.

### 3.4 Training Procedure

The training procedure of ELAS integrates low-rank optimization with activation 2:4 sparsity. ELAS begins with a dense warmup phase, where the model is trained with dense activations for $N_{warmup}$ steps to establish stable representations. Following this, sparse training is applied using 2:4 activation sparsification to all forward passes while maintaining low-rank constraints on weight matrices.

For gradient computation, we use the straight-through estimator for activation gradients and standard backpropagation for low-rank parameters. Optimizer updates are handled by the Riemannian Optimizer that respects orthogonality constraints and supports periodic exact Riemannian updates. The hyperparameters include only dense warmup steps $N_{warmup}$.

This integrated approach enables efficient training by combining parameter reduction through low-rank decomposition with computational acceleration through sparse activations, while maintaining competitive model performance. The structured activation sparsification becomes particularly beneficial for large batch training scenarios, where it significantly reduces memory overhead for activation storage. Algorithm 2 details the complete ELAS training procedure.

## 4 Experiments

We evaluate the performance of ELAS through comprehensive experiments on large language model pre-training. We conducted detailed ablation studies to further demonstrate ELAS's effectiveness. All experiments were performed on NVIDIA 3090/A100 GPU.

### 4.1 Experiments setup

**Dataset.** Our pre-training experiments utilize the Colossal Clean Crawled Corpus (C4) dataset (Raffel et al., 2020). This dataset represents a comprehensive collection of web-scraped text data that undergoes rigorous cleaning and filtering processes, making it a standard choice for language model pre-training tasks. Following the common practice from prior work (Li et al., 2025; Han et al., 2024), we train all models for one complete epoch without repetition.

---

**Algorithm 2:** ELAS

---

1: **Input:** Dataset $\mathcal{D}$, warmup steps $N_{warmup}$, total steps $N_{total}$
2: **Input:** Ranks $r_{attn}$, $r_{mlp}$, refresh frequency $f_{refresh}$
3: **Output:** Trained model parameters $\{\mathbf{A}_i, \mathbf{B}_i\}$
4: Initialize low-rank matrices $\{\mathbf{A}_i, \mathbf{B}_i\}$ with Xavier initialization
5: $step \leftarrow 0$
6: **while** $step < N_{total}$ **do**
7:     Sample batch $(\mathbf{x}, \mathbf{y})$ from $\mathcal{D}$
8:     **Forward Pass:**
9:     **for** each layer $i$ with low-rank weights $\mathbf{W}_i = \mathbf{A}_i \mathbf{B}_i$ **do**
10:         Compute activation: $\mathbf{z}_i = f(\mathbf{W}_i \mathbf{h}_{i-1})$
11:         **if** $step \geq N_{warmup}$ **then**
12:             $\mathbf{z}_i \leftarrow \text{sparsify}_{2:4}(\mathbf{z}_i)$ {Call Algorithm 1}
13:         **end if**
14:     **end for**
15:     Compute loss: $\mathcal{L} = \text{loss}(\text{model}(\mathbf{x}), \mathbf{y})$
16:     **Backward Pass:**
17:     Compute gradients w.r.t. $\{\mathbf{A}_i, \mathbf{B}_i\}$ using STE for sparse activations
18:     **Optimizer Update:**
19:     Apply LORO optimizer updates with orthogonality constraints
20:     **if** $step \bmod f_{refresh} = 0$ **then**
21:         Refresh optimizer state to maintain representational capacity
22:     **end if**
23:     $step \leftarrow step + 1$
24: **end while**

---

**Model Architecture.** We refer to (Touvron et al., 2023) using a llama-based architecture and test models ranging from 60M to 1B parameters. This architecture implementation includes pre-normalization layers, RMSnorm normalization, and the Swiglu activation mechanism (Zhang & Sennrich, 2019; Shazeer, 2020). Our experimental framework follows established methods from recent research and utilizes BF16 precision to improve memory utilization. Specific details of our optimizer settings, cosine annealing learning rate strategy, and warmup procedure are referred to (Li et al., 2025; Zhao et al., 2024). Detailed parameter configurations for each model size are provided in Table 6.

### 4.1.1 Baselines

We benchmark ELAS against several established methods: standard Full-Rank pre-training serves as our primary baseline, along with LoRA (Hu et al., 2021) and other state-of-the-art pre-training techniques, including ReLoRA (Lialin et al., 2023), GaLore (Zhao et al., 2024), LORO (Mo et al., 2025), CoLA (Liu et al., 2025), and SLTrain (Han et al., 2024). All comparisons maintain equivalent training token budgets to ensure fair evaluation.

### 4.1.2 Hyper parameters selection

As we mentioned before, ELAS requires no additional parameter tuning. We implement a dense warmup period before activating 2:4 sparsity training. For the dense warmup steps, we set the dense warmup duration to 1000 steps following (Haziza et al., 2025). We also follow the common pre-training settings, which linearly increase the learning rate before the first 10% of training tokens.

## 4.2 Main results

### 4.2.1 ELAS shows competitive performance with baseline methods

As shown in Table 1, ELAS demonstrates competitive performance across all model sizes while maintaining the memory efficiency benefits of low-rank training. Specifically, ELAS achieves perplexity scores very close to the LORO

Table 1: We report the comparative performance in terms of perplexity, along with parameter statistics (Param, in millions) and memory requirements (G, in gigabytes) for each method. Here, $r$ indicates the target rank while $d$ refers to the model's hidden dimension. The memory consumption includes model parameters and gradients but excludes activations. The reported baseline results are obtained from (Li et al., 2025; Zhao et al., 2024; Han et al., 2024)".

| | 60M | | | 130M | | | 350M | | | 1B | | |
|---|---|---|---|---|---|---|---|---|---|---|---|---|
| $r/d$ | 128/512 | | | 256/768 | | | 256/1024 | | | 512/2048 | | |
| Tokens | 1.1B | | | 2.2B | | | 6.4B | | | 13.1B | | |
| Method | PPL↓ | Param(M) | Mem(G) | PPL↓ | Param(M) | Mem(G) | PPL↓ | Param(M) | Mem(G) | PPL↓ | Param(M) | Mem(G) |
| Full-Rank | 34.06 | 58 | 0.35 | 24.36 | 134 | 0.81 | 18.80 | 368 | 2.21 | 15.56 | 1339 | 8.04 |
| LoRA | 34.99 | 58 | 0.36 | 33.92 | 134 | 0.84 | 25.58 | 368 | 1.85 | 19.21 | 1339 | 6.34 |
| ReLoRA | 37.04 | 58 | 0.36 | 29.37 | 134 | 0.84 | 29.08 | 368 | 1.85 | 18.33 | 1339 | 6.34 |
| GaLore | 34.88 | 58 | 0.28 | 25.36 | 134 | 0.61 | 18.95 | 368 | 1.59 | 15.64 | 1339 | 4.76 |
| CoLA | 34.10 | 43 | 0.24 | 25.61 | 94 | 0.57 | 19.75 | 185 | 1.11 | 15.76 | 609 | 3.66 |
| SLTrain | 34.15 | 44 | 0.26 | 26.04 | 97 | 0.60 | 19.42 | 194 | 1.24 | 16.14 | 646 | 4.16 |
| LORO | 33.87 | 43 | 0.24 | 24.78 | 94 | 0.57 | 19.66 | 185 | 1.11 | 15.53 | 609 | 3.66 |
| **ELAS** | 34.12 | 43 | 0.24 | 24.80 | 94 | 0.57 | 19.94 | 185 | 1.11 | 15.69 | 609 | 3.66 |

Table 2: FFN activation memory consumption comparison between LORO and ELAS for the 1B parameter model across different batch sizes. Sequence length is set to 2048. All values are in GB.

| Method | 1 | 2 | 4 | 8 | 16 | 32 | 64 | 128 |
|---|---|---|---|---|---|---|---|---|
| LORO | 1.42 | 2.84 | 5.68 | 11.36 | 22.71 | 45.43 | 90.85 | 181.71 |
| ELAS | 0.80 | 1.59 | 3.18 | 6.36 | 12.72 | 25.44 | 50.88 | 101.76 |

baseline with only minimal degradation ranging from 0.07 to 0.28 perplexity points. Notably, ELAS maintains the same parameter count and memory footprint as LORO while adding the benefits of activation sparsity for computational acceleration and activation memory reduction, which will be detailed in the next section.

Compared to full-rank training, ELAS achieves competitive performance while using significantly fewer parameters and consuming less memory. When compared to other low-rank methods, ELAS consistently outperforms LoRA and ReLoRA across all model sizes. ELAS achieves comparable results to more sophisticated methods such as GaLore and CoLA. These results validate that applying 2:4 activation sparsity introduces minimal performance overhead while preserving the substantial efficiency gains of low-rank training.

### 4.2.2 ELAS achieves activation memory reduction

ELAS provides substantial memory savings through 2:4 structured sparsity applied to FFN activations. Table 2 presents the activation memory consumption comparison between LORO and ELAS for the FFN module of 1B parameter model across various batch sizes.

The results demonstrate that ELAS consistently achieves a reduction in FFN activation memory consumption regardless of batch size. This memory saving stems from the efficient 2:4 sparse storage format supported by NVIDIA GPUs, which stores only the non-zero values along with compact indexing information. The memory reduction becomes valuable for large-batch training scenarios, where activation memory consumption can become a significant bottleneck during training. This validates the scalability of our approach and its applicability across different training configurations.

### 4.2.3 ELAS aobtains computational acceleration

We evaluate the inference phase computational acceleration of ELAS compared to full-rank across different model sizes and sequence lengths. We focus on the feed-forward network modules where 2:4 activation sparsity is applied. Table 3

Table 3: Computational speedup of ELAS compared to full-rank training across different model sizes and sequence lengths. Values represent the speedup ratio of FFN module (ELAS time / Full-rank time), where values > 1.0 indicate acceleration.

| Model | 512 | 2048 | 4096 | 8192 | 16384 | 32768 | 65536 |
|---|---|---|---|---|---|---|---|
| 60M | 0.50× | 1.57× | 1.75× | 1.50× | 1.55× | 1.59× | 1.55× |
| 130M | 0.75× | 1.56× | 1.80× | 1.52× | 1.51× | 1.53× | 1.52× |
| 350M | 1.29× | 1.87× | 1.86× | 1.82× | 1.85× | 1.88× | 1.88× |
| 1B | 2.05× | 2.47× | 2.48× | 2.55× | 2.63× | 2.73× | 2.75× |

presents the speedup results measured as the ratio of ELAS inference time to full-rank inference time with sequence lengths ranging from 512 to 65,536.

According to the results, we can see that ELAS achieves significant computational acceleration, with speedups ranging from 1.5× to 2.75× for most configurations. Interestingly, the effectiveness of ELAS on computational acceleration is positively correlated with model size and sequence length. Larger models exhibit greater acceleration, with the 1B parameter model achieving the most significant speedups across all sequence lengths.

Notably, shorter sequences (512 tokens) sometimes exhibit performance degradation compared to full-rank, especially with smaller models, i.e., 60M and 130M. This can be attributed to the overhead of kernel initialization and sparse matrix operations, which becomes proportionally smaller as the computational workload increases. Overall, activation 2:4 sparsity can provide significant performance acceleration with larger models and longer sequences.

## 4.3 Ablation study

We conduct two ablation studies to investigate the key components of ELAS, i.e., the activation 2:4 dense warmup phase and the activation 2:4 sparse methods. Due to computational constraints, the first ablation experiments are performed on the 60M and 130M parameter models.

### 4.3.1 Ablation on activation 2:4 dense warmup steps

We conduct the ablation on the impact of dense warmup phase before activating 2:4 activation sparsity. Table 4 presents the final perplexity results for different warmup steps.

The results show the importance of dense warmup for training stability. When no warmup is applied (0 steps), the 60M model completely fails to converge, while the 130M model shows significantly degraded performance. This instability arises because in the early training stages, the natural sparsity of activations after ReLU² is relatively low (around 50%). It makes the 2:4 sparse training prone to significant information loss that disrupts the learning process. Both models achieve improved performance with appropriate warmups where further extending the warmup beyond 1000 steps shows diminishing returns.

These findings confirm that the dense warmup is essential for activation 2:4 sparse training, allowing the model to establish stable optimization dynamics before introducing the structural constraints of activation sparsity.

Table 4: Ablation study on the effect of dense warmup steps before applying activation 2:4 sparsity. Results show final evaluation perplexity for different warmup steps on 60M and 130M models.

| Model | 0 | 500 | 1000 | 2000 | 3000 |
|---|---|---|---|---|---|
| 60M | NaN | 36.71 | 34.12 | 34.48 | 34.32 |
| 130M | 29.7 | 25.76 | 24.80 | 24.93 | 24.83 |

### 4.3.2 Ablation on activation 2:4 sparse methods

We compare different approaches for applying 2:4 structured sparsity to activations. Our default method (naive) directly selects the top-2 elements by magnitude within each group of 4 consecutive values. We also evaluate the soft thresholding approach from S-STE (Hu et al., 2024b), which was originally designed for weight sparsification.

The original S-STE method (soft_weights) subtracts the second-largest weight value from each group of 4 consecutive weights. It then applies scaling to minimize the Frobenius norm difference between the original and sparsified weight matrices. However, there is no support for directly applying this weight-based scaling to activation sparsification, as activations have different statistical properties than weights. To address this mismatch, we propose an adapted version (soft_activation) that computes the scaling factor based on a small batch of input activations (input × weights) rather than using weight statistics. This approach better reflects the true activation distribution when determining the soft thresholding parameters.

Table 5 presents the experimental results across different model sizes. The naive magnitude-based approach achieves the best performance and training stability across all model sizes. The original soft_weights shows performance degradation and training instability, failing to converge for larger models. Our activation-based adaptation Soft_activation performs better than soft_weights but still exhibits instability and generally worse performance than the naive approach.

These results suggest that the soft thresholding designed for weight sparsification do not align well to activation sparsification. The direct magnitude-based selection proves most effective for maintaining both training stability and model performance when applied to activations.

Table 5: Ablation study on different 2:4 sparse methods applied to activations. Results show evaluation perplexity across different model sizes. "NaN" indicates training instability leading to divergence.

| Model | Naive | Soft_weights | Soft_activation |
|---|---|---|---|
| 60M | 34.12 | 39.56 | 34.01 |
| 130M | 24.80 | 27.12 | 25.36 |
| 350M | 19.94 | NaN | 20.55 |
| 1B | 15.69 | NaN | NaN |

## 5 Conclusion

In this paper, we presented ELAS, a novel framework that combines low-rank weight training with 2:4 structured activation sparsity for efficient LLMs pre-training. By applying the LORO framework with ReLU² activation functions and structured sparsity of forward activations, ELAS achieves a balance between training efficiency and model performance.

Our experimental results on LLaMA models ranging from 60M to 1B parameters demonstrate that ELAS maintains competitive performance with minimal degradation compared to dense baselines while providing computational and memory benefits. Through ablation studies, we evaluated the importance of dense warmup for training stability and validated that simple magnitude-based 2:4 sparsity outperforms other methods, such as soft thresholding approaches, when applied to activations.

Table 6: Hyperparameters of the LLaMA model. Training data is specified in tokens.

| Params | Hidden | Intermediate | Heads | Layers | Training Tokens |
|---|---|---|---|---|---|
| 60M | 512 | 1376 | 8 | 8 | 1.3B |
| 130M | 768 | 2048 | 12 | 12 | 2.6B |
| 350M | 1024 | 2736 | 16 | 24 | 6.4B |
| 1B | 2048 | 5461 | 24 | 32 | 13.1B |

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
