# OpenReview forum: "ELAS: Efficient Pre-Training of Low-Rank Large Language Models via 2:4 Activation Sparsity"
_TMLR — Withdrawn by Authors_

### Review · Reviewer_N4VZ · 2026-06-03

**Summary Of Contributions:**

This paper proposes ELAS, a new algorithm for improving the computational efficiency of pre-training large language models. The main idea is to combine low-rank weight parameterization with 2:4 structured sparsity applied to the outputs of squared ReLU activations. By applying sparsity to activations rather than to weights, the method aims to exploit NVIDIA GPUs' hardware support for 2:4 structured sparse computation while avoiding substantial degradation in model performance, i.e., perplexity. The authors evaluate ELAS through pre-training experiments on LLaMA-style models of different sizes and report that the method can reduce activation memory and accelerate feed-forward computations with only minimal degradation in perplexity. The main strength of the paper is the simplicity of the proposed method. The idea is intuitive and easy to implement on top of existing low-rank training methods. The experiments suggest that ELAS can reduce activation memory usage while keeping perplexity close to the low-rank baseline. However, the experimental evidence does not fully support the broader claims of training acceleration and practical end-to-end efficiency.

**Additional Comments:**

I read the entire paper, but to be honest, I have low confidence in it because it's far from my  ​​expertise (I'm a theoretical researcher and not familiar with making improvements in implementation layer).

**Audience:**

Yes

**Audience Explanation:**

Reducing the computational and memory costs of large language model pre-training is an important topic in modern machine learning and is clearly relevant to the TMLR readership.

**Broader Impact Concerns:**

nothing in particular.

**Claims And Evidence:**

No

**Claims Explanation:**

The claims are partially supported, but the evidence is not fully convincing for the broader claims made in the abstract. The experiments suggest that ELAS can reduce FFN activation memory and improve FFN computation speed with only small degradation in perplexity. However, the reported evidence is mostly local to the FFN module.

First, Table 1 reports memory consumption excluding activations, while activation memory reduction is one of the main motivations of ELAS. Table 2 reports FFN activation memory, but it is unclear how much the total peak memory is reduced in overall training. Second, the perplexity is compatible with other methods, but no error-bars are reported. Third, the rank setting in Section 3.1, where $r_{\rm attn}=r_{\rm mlp}=256$, seems inconsistent with Table 1.

For computational efficiency, Table 3 appears to define the speedup ratio incorrectly: it should likely be “Full-rank time / ELAS time,” not “ELAS time / Full-rank time.” Moreover, the reported maximum 2.75x speedup is for the FFN module, not for end-to-end training or inference. End-to-end wall-clock speedups or tokens-per-second would better support the claim of training and inference acceleration.

Finally, the training token counts in Tables 1 and 6 appear inconsistent for the 60M and 130M models, although Section 4.1.1 states that all comparisons use equivalent token budgets. This should be clarified.

**Requested Changes:**

The authors should clarify and strengthen the experimental evidence for the efficiency claims. In particular, they should report end-to-end peak memory usage, since Table 1 excludes activations and Table 2 only reports FFN activation memory. They should also provide uncertainty estimates, such as multiple-seed results or error bars, for the reported perplexities, or at least discuss this limitation.

The authors should clarify the experimental settings. The rank setting in Section 3.1 appears inconsistent with Table 1, and the training token counts in Tables 1 and 6 differ for the 60M and 130M models. These inconsistencies should be corrected.

The computational speedup results also need clarification. The speedup ratio in Table 3 appears to be defined in the wrong direction, and the reported speedups are for the FFN module rather than end-to-end training or inference. The authors should report wall-clock speedups or tokens-per-second for full training and inference, or weaken the corresponding claims in the abstract.

---

### Review · Reviewer_GoGx · 2026-06-16

**Summary Of Contributions:**

The submission presents ELAS, an approach for pre-training low-rank language models with 2:4 structured sparsity applied to activations. The method uses LORO as the underlying low-rank training framework, changes the MLP block from the usual SwiGLU-style structure to a ReLU$^2$ feed-forward layer, and then sparsifies the post-activation tensor by keeping the two largest entries in every group of four. Since this operation is not differentiable, the authors rely on a straight-through estimator during training.

Experiments are conducted on LLaMA-like models with sizes from 60M to 1B parameters using the C4 corpus. The reported results suggest that ELAS stays close to LORO in perplexity, while offering lower FFN activation storage and faster FFN computation in some settings. The paper further studies the role of dense warmup and compares a few variants of the activation sparsification procedure.

I find the problem setting meaningful. Reducing the cost of pre-training is an important goal, and it is reasonable to study whether low-rank training and hardware-supported structured sparsity can be combined. Moving sparsity from weights to activations is also a reasonable design choice, since sparse weight training often introduces optimization and implementation complications.

My main reservation is that the current experiments do not yet establish the full efficiency story. The paper argues for efficient pre-training, but most of the evidence concerns FFN-level memory or timing rather than complete training runs. It is therefore unclear how much practical speedup or peak-memory reduction one should expect in an actual pre-training setting. The method also bundles together several changes, including low-rank parameterization, replacing SwiGLU with ReLU$^2$, applying 2:4 activation sparsity, and using a warmup phase. The current ablations do not cleanly separate these effects. I therefore view the paper as promising, but not yet fully convincing in its current form.

**Additional Comments:**

N/A

**Audience:**

Yes

**Audience Explanation:**

The paper addresses a topic that is likely to interest researchers working on efficient training, sparse neural networks, model compression, and systems-aware language-model design. Training large language models remains expensive, and methods that reduce memory or exploit hardware-supported sparsity are clearly relevant to the community.

The most interesting result is that 2:4 activation sparsity seems compatible with low-rank pre-training, at least up to the 1B scale tested here, without a major perplexity penalty. The ablations also provide useful practical observations. For example, the dense warmup result is a good reminder that sparsity constraints can be harmful early in training, and the failure of some soft-thresholding variants suggests that methods designed for sparse weights may not automatically work for sparse activations.

However, the current version is less compelling as a complete efficiency paper. I think readers would still find the idea and empirical trends useful, but the paper would need stronger end-to-end measurements before it can firmly support its main practical claims.

**Broader Impact Concerns:**

I do not see serious broader-impact concerns specific to this work. The paper is about making language-model pre-training more efficient, which could reduce memory requirements and possibly lower compute cost for a fixed training setup.

At the same time, the paper should be cautious about making environmental or accessibility claims without direct evidence. An efficiency method can reduce the cost of a given experiment, but it can also encourage training more or larger models. A short broader-impact statement acknowledging both possibilities would be sufficient.

**Claims And Evidence:**

No

**Claims Explanation:**

The paper does provide useful evidence for part of its claim. In particular, the perplexity results indicate that adding 2:4 activation sparsity on top of LORO does not cause a large quality drop in the tested settings. The warmup ablation is also helpful: it shows that immediately imposing sparsity can break or substantially hurt training, while a short dense warmup improves stability. The comparison between the naive magnitude-based sparsifier and the soft-thresholding variants is also a useful diagnostic result.

That said, the main claim is broader than what the evidence currently supports. The paper is presented as a method for efficient pre-training, but the reported acceleration is mostly measured at the FFN/module level. This leaves open the central practical question: does the full training pipeline become faster? End-to-end training time may be affected by many other costs, including sparsification overhead, backward computation, optimizer updates, LORO-specific refresh steps, communication or memory movement, and non-FFN layers. Without full training throughput or wall-clock measurements, the efficiency claim remains incomplete.

The memory results are also not fully satisfactory. The main comparison table reports parameter and gradient memory but explicitly leaves out activations. Since the key novelty of the paper is activation sparsity, this makes the main memory comparison less informative. The separate activation-memory table is useful, but it is limited to FFN activations and does not show measured peak GPU memory during actual training. It would be much more convincing to report real peak memory, maximum feasible batch size, or both.

I am also not fully convinced that the baseline comparison is controlled carefully enough. Some baseline numbers appear to come from prior papers, while ELAS is evaluated separately. The paper should make clearer whether all methods use the same model architecture, tokenizer, data processing, number of training tokens, optimization setup, precision, hardware, and evaluation protocol. This matters especially because ELAS changes the activation function and FFN structure, so the comparison is not only about sparsity.

For these reasons, I would say that the current evidence supports the feasibility of the idea, but not the stronger claims about practical pre-training efficiency.

**Requested Changes:**

1. Provide end-to-end training efficiency results.
The paper should report full training-loop measurements, not only FFN-level timing. Useful metrics would include tokens/sec, step time, total wall-clock training time, or achieved throughput under the same hardware setting. Since the paper is framed around efficient pre-training, this is the most important missing evidence.
2. Report practical training-memory savings.
The current memory comparison is hard to interpret because the main table excludes activations, while the proposed method mainly targets activation memory. The authors should report measured peak GPU memory during training, ideally across several batch sizes or model sizes. Showing the maximum feasible batch size under a fixed GPU budget would also make the practical benefit clearer.
3. Add a cleaner ablation separating ReLU$^2$ from 2:4 activation sparsity.
ELAS changes the FFN activation structure and applies activation sparsity at the same time. To isolate the contribution of sparsity, the authors should compare at least: LORO with the original SwiGLU FFN, LORO with ReLU$^2$ but dense activations, and LORO with ReLU$^2$ plus 2:4 sparse activations. This would make the source of the reported gains and losses much clearer.
4. Clarify the fairness of baseline comparisons.
The paper should explicitly state which baseline numbers are reproduced by the authors and which are taken from prior papers. It should also clarify whether the compared methods use the same architecture, tokenizer, data preprocessing, token budget, optimizer, precision, hardware, and evaluation protocol. If some settings differ, the paper should phrase comparative claims more cautiously.
5. Clarify and correct the sparse-computation evaluation.
The authors should explain whether ELAS uses NVIDIA 2:4 sparse tensor cores, custom Triton kernels, masked dense operations, or another implementation. They should also specify whether sparse computation is used during training, inference, or both, and whether the backward pass benefits from sparsity. In addition, Table 3 appears to define the speedup ratio as “ELAS time / Full-rank time” while interpreting values greater than 1 as acceleration; this definition should be corrected.

---

### Review · Reviewer_Hv7d · 2026-06-16

**Summary Of Contributions:**

The paper proposes ELAS, a training framework that combines low-rank LLM pre-training with 2:4 structured activation sparsity. The method builds on LORO-style low-rank weight training, replaces the standard LLaMA MLP with a simpler ReLU² FFN, and then applies magnitude-based 2:4 sparsification to the FFN activations. Gradients through the sparsification step are handled with a straight-through estimator.

The main idea is straightforward, low-rank weights reduce parameter and optimizer memory, while 2:4 activation sparsity reduces FFN activation memory and can use hardware-supported sparse matrix multiplication. The paper evaluates this on LLaMA-style models from 60M to 1B parameters trained on C4, compares perplexity against full-rank and low-rank baselines, reports FFN activation-memory savings for the 1B model, gives FFN-module speedups across model sizes and sequence lengths, and includes two small ablations on warmup length and activation sparsification strategy.

### Strengths:
1. The problem is real. Activation memory and FFN compute can become meaningful bottlenecks in large-batch or long-context training, and it is reasonable to look beyond weight sparsity if weight sparsification hurts accuracy. The paper’s focus on activation sparsity inside low-rank pre-training is a practical angle.
2. The method is simple and hardware-aware. ELAS is not an overcomplicated system: it uses ReLU² to induce sparse activations, keeps the top-2 entries in each group of 4, and trains with STE. That simplicity is a plus if the actual speedups hold end to end.
3.  The reported perplexity degradation relative to LORO is small. In Table 1, ELAS is close to LORO across 60M–1B settings, while keeping the same parameter count and parameter/gradient memory numbers as LORO.

### Weaknesses:
1.  The paper’s strongest claims are not supported cleanly. It talks about efficient pre-training and training acceleration, but the clearest speedup table is for FFN-module inference time, not end-to-end pre-training wall-clock. That is a serious gap for a paper whose title and pitch are about pre-training efficiency.
2.  The method is confounded by an architecture change. ELAS does not just add activation sparsity. It also replaces SwiGLU with ReLU² and simplifies the MLP. The paper does not sufficiently isolate how much of the result comes from low-rank training, ReLU², removed gating, or 2:4 sparsity.
3.  The evaluation is narrower than the writing suggests. The experiments are on C4 perplexity up to 1B parameters, with memory tables that exclude activations in the main comparison and a separate FFN-only activation-memory table. There is no convincing end-to-end training throughput table, no downstream evaluation, and no full accounting of total memory including activations.

**Audience:**

Yes

**Audience Explanation:**

Efficient LLM pre-training is a topic many TMLR readers care about. A simple recipe that combines low-rank training with hardware-friendly activation sparsity would be useful if it really delivers practical training savings with little quality loss. The paper is especially relevant to people working on low-rank optimization, structured sparsity, sparse kernels, and memory-efficient transformer training.

That said, the interest is mostly practical rather than conceptual. This is not a deep theoretical paper and not a new model family. The paper’s value lives or dies on whether the system gives real, reproducible end-to-end savings. Right now, the paper gives some promising module-level and memory numbers, but not enough to settle that question.

**Broader Impact Concerns:**

I do not see major ethical concerns requiring special review. The paper is about efficient pre-training, and the main societal angle is standard. If effective, it could reduce compute cost and energy use for LLM training, but it could also make large-scale model training more accessible to actors with less oversight

**Claims And Evidence:**

No

**Claims Explanation:**

The paper provides evidence that ELAS can preserve perplexity reasonably well and reduce FFN activation memory under the authors’ setup. It also provides evidence that sparse FFN computation can be faster in some module-level inference configurations. But the paper does not convincingly support the broader claim of efficient LLM pre-training with practical training acceleration, because the most important measurements are missing or too narrow.

**Requested Changes:**

### Critical for acceptance

1. The paper is about efficient pre-training, so it must report wall-clock training tokens/sec or samples/sec for full models, including forward, backward, optimizer update, attention layers, sparse activation overhead, and dense warmup. FFN-module inference speed is not enough. This is the single most important missing experiment.
2. Table 1 excludes activations, and Table 2 only reports FFN activation memory for the 1B model. The authors should provide a full peak-memory breakdown, including parameters, gradients, optimizer state, activations, sparse metadata, and temporary buffers. Otherwise the memory claim is incomplete.
3. The method changes the architecture and sparsifies activations at the same time. Please add ablations such as LORO + ReLU² dense, ELAS without sparsity, full-rank + ReLU², and ideally LORO + standard SwiGLU. Without this, the paper cannot say clearly what ELAS itself contributes.
4.  The table caption says values are “ELAS time / Full-rank time” and values above 1 indicate acceleration. That ratio would normally mean ELAS is slower, not faster. The authors likely mean “Full-rank time / ELAS time.” This must be fixed because it affects the main efficiency claim.

### Would strengthen the work
1. Add downstream or held-out evaluations beyond C4 perplexity.
2. Add results beyond 1B parameters if possible, or at least explain why 1B is enough evidence for the intended scaling claim.
3. Add sensitivity to warmup length and sparsity activation timing for larger models as well, not only 60M and 130M.

---

### Note · Authors · 2026-06-25

I have read and agree with the venue's withdrawal policy on behalf of myself and my co-authors.